EDUCATION

# Ten rules for a structural bioinformatic analysis

Stephanie A. Wankowicz [1,2]*

**1** Departments of Molecular Physiology and Biophysics, Biochemistry, Computer Science, Vanderbilt University, Nashville, Tennessee, United States of America, **2** Center for Applied AI in Protein Dynamics, Center for Structural Biology, Vanderbilt University, Nashville, Tennessee, United States of America

* stephanie@wankowiczlab.com

## Abstract

The Protein Data Bank (PDB) is one of the richest open-source repositories in biology, housing over 242,000 macromolecular structural models alongside much of the experimental data that underpins these models. By systematically collecting, validating, and indexing these models, the PDB has accelerated structural biology discoveries, enabling researchers to compare new entries against a vast archive of solved structures and, more recently, powering protein structure prediction. Leveraging this wealth of data, structural bioinformatics has uncovered patterns, such as conserved protein folds, binding-site features, or subtle conformational shifts among related proteins, that would be impossible to detect from any single structure. Through the democratization of structural data and open-source analytical tools, now amplified by the power of large language models, a broader community of researchers is equipped to drive new scientific discoveries using structural data. However, good structural bioinformatics requires understanding some of the nuances of the underlying experimental data, data encoding conventions, and quality control metrics that can affect a model's precision, fit-to-data, and comparability. This knowledge, combined with developing good controls, statistics, and connections to other databases, is essential for drawing accurate and reliable conclusions from PDB data. Here, we outline 10 recommendations for doing structural bioinformatic analyses crafted to pave the way for others to uncover exciting discoveries.

## Author summary

Here, we provide a roadmap for users to leverage the Protein Data Bank's vast collection of protein structural models into reliable and valuable insights. It lays out 10 clear rules that help readers quality control their data, choose fair comparison sets, and judge model quality so results aren't led astray by noise, bias, or overconfidence. The guide also shows how to connect structures to other databases. By highlighting best practices, such as utilizing re-refined models and being aware of common pitfalls, we guide users to leverage this rich data

**Data availability statement:** No data involved.

**Funding:** The author(s) received no specific funding for this work.

**Competing interests:** The author has declared that no competing interests exist.

for enhanced biological insights. These guidelines will enable stronger, more re-producible structural analyses that accelerate drug discovery, illuminate disease mechanisms, and make open data broadly useful across the life sciences.

## Introduction

In 1971, with only seven structures, the first, and still active, open-access database in biology was created: the Protein Data Bank (PDB) [1]. From these modest beginnings, the PDB has grown to include over 242,000 structures [2], and its systematic archiving of macromolecular models has reshaped the field, transforming our understanding of the relationship between structure and biological function and enabling advances from elucidating enzyme catalysis to the rational design of new therapeutics. Critically, when the Research Collaboratory for Structural Bioinformatics (RCSB) PDB was established, one of their first major undertakings was a large-scale remediation of legacy data, addressing inconsistent formats, incomplete metadata, and nonstandard nomenclature to enable systematic analysis and ensure that the archive could support future large-scale structural bioinformatics. The remediation effort, later extended by the wwPDB partners (PDBe, PDBj, and Biological Magnetic Resonance Bank [BMRB]), standardized chemical components, corrected errors, and transitioned the archive to the current, more robust mmCIF format [2–4]. These efforts laid the foundation for structural bioinformatics by ensuring that PDB data were reliable, interoperable, and machine-readable.

Today, the PDB is maintained as a single, global archive through the Worldwide Protein Data Bank (wwPDB) consortium, which coordinates deposition, validation, and dissemination of macromolecular structures. The consortium comprises regional data centers, RCSB PDB in the United States, PDBe in Europe, PDBj in Japan, each providing unique portals, visualization tools, and database integrations tailored to their respective communities [5–7]. All sites share a unified deposition system, ensuring that structures are consistently validated and mirrored worldwide within 24 hours of release [8]. In addition, the Electron Microscopy Data Bank (EMDB), jointly maintained with the PDB, serves as the central repository for cryo-electron microscopy (cryo-EM) electron potential maps, enabling joint deposition of maps and models. Together, these interconnected entities provide a comprehensive and interoperable ecosystem that continues to accelerate discovery across the structural biology community.

The original vision of the PDB centered on the deposition of individual structures, where each entry told a story about a protein's fold, function, or interaction. This one-structure, one-story mode of structural biology has yielded an enormous wealth of knowledge and fundamentally shaped our understanding of biology. But one of the greatest strengths of the PDB lies in its ability to uncover new biological insights beyond the scope of a single structure. This has spurred the field of structural bio-informatics, which, through examining patterns over tens to thousands of structures, has identified relationships of protein families and folds, the role of evolution

driving protein structure and function, and information on macromolecular interactions and catalysis [9–16]. Structural bioinformatics also created the critical foundation for the protein structure prediction breakthrough [17–21]. By analyzing thousands of structures at once, you gain the statistical power to detect subtle variations that, when aggregated, have the potential to reveal robust patterns in allostery, ligand binding, macromolecular assembly, and catalysis [9,12]. These analyses can be incredibly powerful alone or in conjunction with other bioinformatic databases, prospective experiments, or theoretical models.

At the core of every PDB structural model lies the atom table, which records the atomic coordinates along with key attributes such as atom type, residue identity, B-factor (atomic displacement parameter), and occupancy. The adoption of the mmCIF format has provided a far richer and more extensible representation than the legacy PDB format. Unlike the fixed-column limitations of PDB files, mmCIF can accommodate the growth of structural biology, including new ligands with five-character identifiers and very large macromolecular assemblies that exceed the capacity of the original format [22]. Another key advantage of mmCIF is its ability to map these deposited coordinates to canonical protein sequences, enabling seamless integration with UniProt and related databases [23]. mmCIF also underpins emerging resources such as PDB-IHM [24], which supports the deposition of integrative and ensemble models, and it provides the extensibility needed for new schemas, including recent work on hierarchical representations of conformational and chemical heterogeneity [25].

The democratization of coding skills, facilitated by large language models, has enabled more users to delve into structural bioinformatics, build hypotheses, support experimental findings, or make independent discoveries. However, structural data is nuanced and can be challenging to work with; without proper quality control or control analyses, it can lead to inaccurate conclusions. Users should understand limitations, potential pitfalls, and caveats to use the data to its full potential.

The guidelines outlined here stem from the lessons and challenges I and others have encountered while performing structural bioinformatics projects, with many of the lessons being applicable for machine learning applications as well. Although predicted structures are increasingly valuable in bioinformatics research [26–28], this article emphasizes experimentally derived structures. While each rule is not universally applicable, consider each recommendation carefully and evaluate how it may relate or be tweaked to fit their problem.

So, first things first, what question do you want to ask, or hypothesis do you want to test? For example, are you looking at the overall protein fold, or does your question require you to know the rotamer angles and only look at wild-type structures of a specific protein? Knowing the answers to these questions is critical for determining your selection criteria, statistical power, analysis, and controls, as outlined below.

## Recommendation #1: Define your biological selection criteria

When starting a structural bioinformatics project, the first step is to define the biological criteria for your study. Consider the structures you need to answer your research question, whether it involves all lysozymes, a specific tyrosine kinase, or all enzymes. Additionally, you may want to further refine your dataset based on ligands. Small molecules such as glycerol or DMSO are often crystallographic additives, while other molecules may be native or synthetic ligands, leading to differences in how you want to classify each structure. Further, assess whether your protein is part of large complexes by examining the identities of other chains. Identical chains typically reflect symmetry-related protomers, whereas distinct macromolecules reveal a multi-protein complex (see more details in Recommendation #4).

Beyond structural selection, sequence-level considerations remove redundancy and drive clustering and alignment analyses. A significant fraction of PDB entries corresponds to homologous proteins or multiple structures of the same protein. Depending on your question, you may want to filter based on sequence or structure. You can cluster by sequence, using MMseq or CDHit [29,30], or based on structure, using TM-score and CATH [31,32], selecting representatives from each cluster for your downstream analysis based on resolution and R-factors or other metrics (see more

in Recommendation #2). Tools like the PISCES server automate this by removing sequences above a chosen identity threshold and keeping the highest-quality structure from each group [33].

The RSCB PDB offers precomputed sequence clusters at certain thresholds (100–30%), based on the MMseqs2 algorithm [30], which applies variable identity thresholds based on modeled residues. Note that this does not include unmodeled residues, often in terminal or loop regions, which can impact this analysis. For sequence alignments, the PDBe supports multiple sequence alignment (MSA) using Clustal Omega [34] and allows retrieval of FASTA sequences for custom alignment. By leveraging the SIFTS database [35], you can map PDB entries onto CATH or SCOP structural hierarchies, UniProt sequence records, and select structures by fold, superfamily, or sequence-based functional annotation (see more in Recommendation #9) [23,31,36]. Additionally, you can use structural alignments, such as those performed with FATCAT, TM-align, CE, or Smith-Waterman 3D alignment, to provide insights into sequence and structural relationships [37–41]. These can be powerful in identifying similar shapes of proteins, yet with sequence differences. Many of these selections can be made using one of the three PDB APIs [5,7,42].

## Recommendation #2: Determine how you will quality control your data

Beyond determining the biological and sequence selection criteria, it is crucial to consider the experimental data underlying structures to ensure a quality dataset. This begins with identifying the methods of structure determination, such as X-ray crystallography, cryo-EM, nuclear magnetic resonance (NMR), or neutron diffraction. Additional factors include resolution (not applicable for NMR), agreement with structure determination data, and stereochemical accuracy.

Resolution, the most common criterion for structural bioinformatics analysis, sets the theoretical limit on the precision of the structural model and is reported for all structures. High resolution, better than 2.5 Å, is essential for accurate side chain positioning, whereas lower resolution models can still yield valuable insights into overall fold and backbone conformation. In cryo-EM, however, resolution is estimated differently than in crystallography: it is typically calculated using the Fourier Shell Correlation (FSC) between two independently reconstructed half-maps [43]. The FSC curve reflects the degree of agreement between the two maps as a function of spatial frequency, and the resolution is conventionally reported at the point where the correlation falls below a given threshold (commonly 0.143). However, the FSC is not a direct measure of atomic detail in the same way that crystallographic resolution is, but rather a measure of the global similarity between two noisy reconstructions. Complicating matters further, cryo-EM maps often exhibit substantial variation in local resolution across the structure, meaning a single global resolution metric may not faithfully capture the interpretability of all regions of the map [44]. It is also important to note that in both X-ray and cryo-EM, while two structures can have the same resolution, they can be modeled to different levels of accuracy, necessitating the exploration of other metrics.

The PDB publishes global validation metrics, including knowledge-based assessments of atomic models, evaluations of the underlying experimental data, and measures of agreement between model and data, and provides detailed validation reports for each entry, following standards established by the X-ray, NMR, and cryo-EM validation task forces [45–49]. Even at high resolution, nearly all structures have a few local errors, but at lower resolutions, errors become more widespread. As structural models can vary widely in quality, these validation metrics are important to consider maintaining scientific reliability and to minimize the risk of errors propagating into biological interpretation, drug design, or computational modeling.

Geometric metrics, such as Ramachandran outliers, are derived from tools such as MolProbity and PROCHECK [50–52]. In X-ray crystallography, *R*-values quantify the agreement between model and data, with higher values reflecting poorer fits; a value near or above 0.3 is commonly used as a threshold for poor quality [53]. Further, depending on your question, you also may want to examine geometry or fit to real space data of individual residues using tools such as MolProbity, Ringer, or real space correlation coefficients [52,54–57].

In cryo-EM, there is an expanding set of approaches being introduced to evaluate the quality of deposited maps and models [58]. As discussed above, global measures such as FSC between half-maps remain the gold standard for

estimating overall map resolution, while model–map FSC curves assess the consistency between the atomic model and experimental maps [47,59]. Increasingly, local validation has become critical. EMRinger evaluates the accuracy of side-chain placement by comparing electron potential peaks with expected rotamer positions, and Q-scores quantify how well the electron potential supports individual atoms and residues [60,61].

You should also be aware of unmodeled regions, often loops or termini. These can be identified by manually comparing the FASTA sequence, representing the input construct, against the sequence in the PDB structural model or with tools like Seqatoms [62]. One may decide to exclude proteins with large missing segments or model in missing loops (see Recommendation #3). The PDB-REDO team has developed an algorithm (Loopwhole) to help fill in many of these missing loops, but it is most effective when high-quality homologous structures are available and when the experimental electron density supports accurate grafting and refinement [63]. Models that include filled loops will be classified as "rebuilt" in the PDB-REDO database. If you include structures with unresolved regions, acknowledge this limitation and adjust your analysis accordingly (see more in Recommendation #8).

Beyond proteins, structures often include small molecules, nucleic acids, carbohydrates, or other molecules of varying quality [64]. Small molecule ligand quality is assessed by agreement with experimental data and geometric accuracy [65], with the latter being evaluated against Cambridge Structural Database reference structures [66]. Metals are also checked by CheckMyMetal, which evaluates metal coordination geometry, bond valency, and potential steric clashes [67]. For nucleic acids, PDB-REDO has introduced validation routines to assess the normality of Watson–Crick base-pair geometry, while DNATCO provides complementary validation of DNA and RNA backbone conformations [68,69].

Ultimately, determining the appropriate experimental selection criteria depends on your research question. For instance, if your research focuses on side chain positioning, higher resolution, lower *R*-values, and precise stereochemical validation are critical. Alternatively, if you are looking for information on the overall protein fold, a broader selection of structures may be acceptable.

## Recommendation #3: Re-processing structural model data

Most structural bioinformatic approaches take information directly from coordinate (PDBx/mmCIF) files. By taking information directly from the coordinate files, you are taking on any errors or biases the original modelers had. Where possible, it is recommended to use X-ray structural models from PDB-REDO [70–72], which reanalyzed the majority of structures in the PDB with experimental data (structure factors), providing uniform automated re-refinement, combined with structure validation and difference-density peak analysis. Since the deposition of reflection data was only encouraged beginning in 1998 and became mandatory in 2008, older structures, whose experimental data were less frequently archived in the PDB, are underrepresented in PDB-REDO [73]. While many models without experimental data can be informative, they come with caveats due to different and older data processing pipelines.

It is also possible to re-process all structures yourself [74,75]. If you are new to refinement, there are many tutorials to get you started [76,77]. Re-processing data can ensure that experimental data is processed in the same way, or allow the application of specific modeling modality or tooling within a refinement program, such as multiconformer modeling, ensemble refinement, 3D variation analysis, or quantum refinement [78–80]. To be able to reprocess your data, you need experimental data to be available, such as MTZ files for X-ray crystallography, maps, half maps, or particle stacks for cryo-EM from the EMDB [81,82], or raw NMR data from the BMRB [81]. After re-processing, similar quality control metrics, as described in Recommendation #2, should be used to evaluate structures.

## Recommendation #4: The PDB and structural models are weird and biased

The PDB is not a uniform sample of all proteins. Because high-resolution crystallography, which comprises the majority of the PDB, favors small, globular, soluble proteins, membrane and flexible or disordered proteins account for roughly 20%–30% of genes but make up less than 2% of PDB entries [83]. Moreover, publication bias further distorts the

distribution of structural models with drug targets, enzymes, and other high-value human proteins accounting for a disproportionate share of PDB entries. This skewing of many structures of the same protein is becoming even more pronounced with an increase in fragment-screening campaigns [84]. As a result, certain protein families dominate the PDB, artificially amplifying their characteristic features in any global analysis. You must consider these redundancies in your analysis, as discussed in Recommendation #2.

In addition to redundancy, it is also important to understand what structural unit is represented in a PDB file. The database distinguishes between the asymmetric unit, the crystallographic unit directly observed in the experiment, and the biological assembly, which represents the functional quaternary structure in vivo. The PDB provides separate mmCIF files for biological assemblies, which are either specified by the authors or inferred computationally by tools such as PISA [85]. For most biological analyses, the biological assembly is the appropriate choice, though it should be noted that approximately 20% of these assemblies may be incorrect, with ProtCID and ProtCAD databases being valuable for sorting true assemblies from crystallographic artifacts [86,87].

Beyond the bias of what structural models exist in the PDB, structural models can be odd and biased. First, it is important to remember that PDB models are just models. They do not explain all the underlying experimental data and can vary depending on the processing pipeline (see Recommendation #3). For example, in X-ray crystallography, crystal contacts, nonbiological interactions between symmetry-related molecules within the crystal lattice, can artificially stabilize particular conformations or create interfaces that don't exist in solution, potentially skewing structural bioinformatics analyses of protein dynamics, flexibility, and genuine interaction sites. We also previously showed that binding site residues are often better modeled than residues outside the binding site [88]. Further, regions of unmodeled residues can arise for many reasons, including resolution and subjective modeling, but automated refinement pipelines cannot correct all of them. All of these issues can lead to structures having different biases. In addition, structures often include unmodeled blobs, frequently ligands.

Finally, all structural data contains extensive conformational and compositional heterogeneity modeled with varying accuracy and encoding [25]. These include anisotropic B-factors, alternative atom locations (altlocs), or multiple models. Anisotropic B-factors describe the direction and magnitude of atomic displacement, while alternative atom locations (altlocs) represent multiple conformations modeled for a single atom [78]. Multiple models, often used in ensemble structures, provide different plausible conformations that together capture the underlying structural variability [89]. While there are ways to encode some of these metrics more uniformly, some encodings cannot be interchanged. Additionally, most bioinformatics libraries, including Biopython, strip out much of this encoding, potentially introducing biases into downstream analyses [90]. To guard against these biases, it is essential to document any data exclusions or alterations made to the data, ensuring accurate comparisons downstream.

## Recommendation #5: Consider your analysis's sample size, statistics, overfitting, and uncertainty

After dataset selection and quality control comes the fun part, looking at and identifying what drives differences between structures. Descriptive bioinformatic analyses, such as cataloguing residue types and counts within binding pockets, are straightforward, but any comparative study requires careful attention to sample size and statistical power. Smaller groups demand larger effect sizes to achieve significance, and paired comparisons should employ paired statistical tests to account for within-pair correlations. Equally important is judging whether observed differences, such as shifts in binding site residue rotamers or altered pocket volumes, are biologically meaningful [91].

When comparing two unpaired groups, choose parametric or nonparametric tests based on data distribution. Parametric tests assume normality, while nonparametric tests are more flexible when distributions are skewed (e.g., residue B-factor values or pocket volumes). For paired data, for example, wild-type vs. mutant, or bound vs. unbound structures, use paired t-tests or Wilcoxon signed-rank tests. Further, be wary of multiple hypothesis testing. Consider adjusting p-values using Bonferroni or false discovery rate corrections. You can also use resampling methods such as jackknife,

bootstrap, or cross-validation to help estimate variability and confidence intervals. Applying well-chosen controls helps guard against false positives and ensures that your findings reflect genuine structural phenomena rather than quirks of a particular dataset.

Avoiding overfitting is equally critical, whether you are working in bioinformatics or machine learning. Where possible, never develop and validate hypotheses on the same data without independent testing. Splitting your dataset into train, test, and validation sets, or employing k-fold cross-validation, is even recommended when defining new structural descriptors or clustering algorithms. Further, consider how you partition your test set, whether by sequence similarity, structural features, or other criteria, to avoid overfitting or memorization [92].

## Recommendation #6: Determine and apply the correct controls

Choosing the proper controls is one of a bioinformatic study's most challenging and often overlooked aspects. Fortunately, the abundance of publicly available structural data makes incorporating negative and positive controls feasible. Controls must directly address the null hypothesis you wish to reject. Negative control datasets, where no effect is expected, are usually easier to define, while positive control datasets, datasets known to exhibit the effect, can be harder to assemble. For example, if you're testing whether a novel structural motif alters protein function, you might compare your proteins of interest against a set of homologous structures that lack the motif. Differences that persist between the groups are more likely to stem from the motif than background variation. You can also randomize specific features, such as residue type or solvent exposure, to break genuine signals, or selectively choose structures that should not display the phenomenon under study [12]. This strategy ensures that any detected signal isn't merely an artifact of the overall distribution of structural features.

For example, consider a case where you argue that hydrophobic residues in binding sites are inherently less dynamic. Alternative explanations might include differences in solvent exposure, secondary-structure context, or biases introduced by your dataset (for instance, selecting only certain CATH classes or ligand types). A robust negative control would examine hydrophobic residues outside binding pockets matched for solvent accessibility and local secondary structure. While it may be impossible to control every variable perfectly, assessing your metric across complementary subsets is critical for demonstrating that your findings reflect genuine biological effects rather than quirks of data selection.

## Recommendation #7: Understand how metrics are compared across your structures

Without careful evaluation, comparison metrics can lead to incorrect conclusions. For instance, larger proteins naturally exhibit higher overall root mean squared distance (RMSD) values, a common metric for comparing the two structures' similarities. Normalizing RMSD by sequence length or reporting RMSD per residue can correct this. Many structure alignment tools, including DALI and TM-align, provide Z-scores indicating the likelihood that an observed similarity would occur by chance [32,93]. Alignment and comparison in torsion space also provide a powerful way to distinguish functionally relevant conformational states. Torsion-angle-based approaches preserve subtle, biologically meaningful differences that are often obscured in atomic coordinate space [94,95].

B-factors, also called temperature factors, atomic displacement parameters, or Debye–Waller factors, estimate each atom's displacement parameter, combining thermal motion of the atom with static disorder from the crystal lattice [96]. Because they arise from the refinement process, B-factors are influenced by data resolution, model bias, occupancy, and lattice packing. As a result, high B-factors do not necessarily guarantee high flexibility in solution. To use them reliably, it's best to normalize B-factors, for example, by Z-scoring within a structure, comparing structures with very similar crystallographic parameters [97,98], and, when possible, corroborating with another metric of flexibility. There are a plethora of other comparison metrics that can be used to compare groups or pairs of PDBs [93,99–101]. Understanding how these metrics are derived and how best to apply them to your analysis is essential to ensuring you use them properly and avoid introducing bias.

## Recommendation #8: Appropriately connect and compare structures

When comparing two groups of structures, it is crucial to balance confounding variables to ensure that biological differences, rather than methodological or crystallographic artifacts, drive the observed differences. Differences in resolution, space group, unit-cell parameters, data processing, and data collection parameters can lead to incorrect conclusions. Depending on the question, this can also include differences in local metrics such as MolProbity or validation scores [102]. Even reprocessing identical raw data with identical refinement settings can yield subtly different models due to stochasticity built into those processes to help with the complex refinement optimization process [74,75]. To minimize such artifacts, applying consistent processing pipelines (such as PDB-REDO) and, where possible, matching crystallographic parameters is important.

These controls become even more critical when looking at pairs of structures, such as ligand-bound versus apo or mutant versus wild type. In these analyses, you often look for subtle conformational changes you want to ensure are not driven by nonbiological artifacts. We recommend pairing structures based on biological differences and ensuring that they have similar crystallographic properties. Some general guidelines include using datasets with resolutions within 0.3 Å, identical space groups, and unit cell dimensions that differ by no more than 10%. While these criteria are not always achievable, deviations can introduce artifacts: differences in crystal contacts or solvent volume may affect the conclusions you can draw.

In some cases, it is valuable to collect structures with diverse crystallographic properties from the same or closely related proteins. Such comparisons can provide insight into conformational heterogeneity and, in particular, are useful for studying loop conformations that crystal contacts may influence. By grouping structures into distinct crystal forms, one can analyze loop conformations across different crystallographic contexts and disentangle genuine biological flexibility from artifacts introduced during crystallization [103,104].

Additionally, you must determine how you will compare structures across groups for all comparisons. For most comparisons, you will need to align structures, often based on the alpha carbon; however, other options include aligning the entire structure or taking sequence into account. Global metrics, such as RMSD, allow you to ignore sequence or small length differences, but if you want to compare specific sections of the protein or amino acids, this will take more care and thought. For example, you may want to compare how a specific loop compares among homologs. This will require aligning structures around that loop or to all residues besides the loop, and also ensuring that crystal contacts are not driving these conclusions.

Comparing structures of the same protein, you can compare using chain and residue IDs, but a standard numbering scheme is required. This can be done by manually renumbering chains and residues or employing algorithms such as PDBRenum to map PDB residue numbers onto UniProt numbering, which also allows for integration with other databases (see Recommendation #9) [105]. If PDBs are similar, you can also align them based on a MSA. One thing to note is that while the MSA will enable renumbering, a single residue number may still correspond to different residue types.

Additionally, it is worthwhile to see if existing databases or collections have the comparisons you want. For example, multiple databases pair apo-holo structures together, although depending on your question, you may want to further curate this database down based on crystallographic properties [106].

## Recommendation #9: Connect your analysis to other databases or prospective experiments

By connecting PDB structures with other bioinformatics databases, you can enrich your analyses with sequence features, domain architectures, pathway contexts, and chemical insights, uncovering deeper relationships between structure, function, and activity. The PDBe API provides programmatic access to sequence, taxonomy, and functional annotations [5]. Family and domain classifications, including Pfam, SCOP, ECOD, and CATH [14,31,35,107,108], are accessible via SIFTS [35]. SIFTS also offers residue-level mappings between PDB structures and UniProt sequences, enabling the labeling of functional sites onto PDB structures [109]. This facilitates comparative analyses, such as examining

conformational changes across a family or correlating structural motifs with functional annotations from Gene Ontology or InterPro [110,111]. PDBs can be connected to pathway and chemical databases such as KEGG and Reactome via UniProt [112,113]. PDBe-KB further consolidates annotations from multiple specialist resources, providing an integrated knowledge base that highlights functional and biological insights mapped onto PDB entries [114]. In addition, the 3D-Beacons network connects structural biology resources across multiple providers, ensuring consistent and federated access to experimental and computational models [115]. While these resources are highly complementary, they are not entirely overlapping, as each database captures different aspects of biological knowledge, and careful integration is often necessary to avoid redundancy or misinterpretation.

Additionally, many PDB structural models have small molecules. PDBe provides excellent ligand pages and tools for analysis within the database [116,117]. Additionally, small molecule information can be linked to existing databases. The PDB's Chemical Component Dictionary assigns ligand IDs that can be cross-referenced with ChEMBL, PubChem, or DrugBank [118–120]. Additionally, external databases such as PDBBind and BindingDB can group chemical or binding information and link it back to PDB information [121,122]. These databases enable easier retrieval of assay data, clinical information, or physicochemical properties of ligands. A growing number of 'curated' databases also look at protein-ligand interactions, post-translational modification, nucleic acid interaction sites, among many others [123–127]. You can then use the pre-calculated metrics or the curated PDB list to calculate the metrics you are interested in.

Additionally, bioinformatics can serve as an excellent partner for hypothesis generation or for supporting prospective experiments. For example, structural bioinformatics can pinpoint the specific residue(s) to mutate to test a desired functional effect, or evaluate whether an experimentally derived hypothesis, such as a loop–domain interaction, holds across homologous structures and influences protein activity.

## Recommendation #10: Visualize everything!

One of the best things about structural biology is visualizing what you are discovering. Looking at structures and the metrics you are using via Pymol or Chimera is a powerful quality control tool for your bioinformatic analyses [128,129]. For example, calculating the comparison between two structures and then manually exploring the metric in a visualization software for a given metric. You can ask: Are you aligning the structures or residues correctly? Does the quantification of the metric you are getting make sense? Once you have confirmed that metrics are calculated correctly and you have results you want to show, Pymol, Chimera, or Coot offer various representations for pieces of the molecule, underlying experimental data, and distance measurements [128–130]. PyMOL can also load molecular dynamics trajectories to visualize conformational changes. ChimeraX's plugin infrastructure efficiently handles larger structures.

## Discussion

Structural bioinformatics provides a robust framework for identifying patterns in macromolecular structures, integrating with other databases, supporting theoretical approaches, and informing prospective experiments [9,12,131]. For example, overlaying quantitative proteomics and large-scale sequence variation onto structural clusters enables identifying regulatory hotspots and prioritizing functionally relevant variants. Additionally, structural bioinformatics can be incredibly powerful in supporting or refuting hypotheses from prospective experiments. While we did not focus on this, AlphaFold or other structure models can help fill gaps where experimental structures are absent [18–20], including now expanding beyond proteins [18,132]. However, users must remain mindful of the "last-Ångstrom" problem, where these prediction models are often inaccurate in very precise measurement, including molecular interactions, residue networks, and the lack of conformational ensembles stemming from these predicted structures [133,134].

Beyond single-structure analyses, statistical and integrative structural biology approaches can help merge structural models to detect new or more subtle changes in structures or structural ensembles. Further, while most of this article focused on how to detect subtle differences using bioinformatics, these tools can be used to go the other way spatially by

integrating cell-scale data to construct multiscale assemblies in their native contexts. We bridge atomistic observations to emergent cellular behaviors, closing the loop between structure, function, and phenotype [135].

Finally, many concepts presented in this paper should also be considered when doing machine learning on protein structures. While protein structure prediction has led to an explosion of machine learning algorithms and approaches applied to structural data, many issues that hinder bioinformatic analyses also arise when splitting datasets in machine learning [92,136,137]. In particular, researchers must carefully avoid information leakage by ensuring that homologous proteins, redundant structures, or closely related crystal forms are not distributed across training and test sets, as this can lead to overly optimistic performance estimates. Incorporating these principles into structural bioinformatics ensures that computational results remain reliable, reproducible, and ultimately informative for guiding experimental design.

## Author contributions

**Conceptualization:** Stephanie A. Wankowicz.

**Data curation:** Stephanie A. Wankowicz.

**Formal analysis:** Stephanie A. Wankowicz.

**Investigation:** Stephanie A. Wankowicz.

**Methodology:** Stephanie A. Wankowicz.

**Project administration:** Stephanie A. Wankowicz.

**Resources:** Stephanie A. Wankowicz.

**Software:** Stephanie A. Wankowicz.

**Validation:** Stephanie A. Wankowicz.

**Visualization:** Stephanie A. Wankowicz.

**Writing – original draft:** Stephanie A. Wankowicz.

**Writing – review & editing:** Stephanie A. Wankowicz.

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
