## [Decision Letter · Decision Letter 0]

14 Aug 2025

10 Rules for a Structural Bioinformatic Analysis

PLOS Computational Biology

Dear Dr. Wankowicz,

Thank you for submitting your manuscript to PLOS Computational Biology. After careful consideration, we feel that it has merit but does not fully meet PLOS Computational Biology's publication criteria as it currently stands. Therefore, we invite you to submit a revised version of the manuscript that addresses the points raised during the review process.

Please submit your revised manuscript within 60 days Oct 14 2025 11:59PM. If you will need more time than this to complete your revisions, please reply to this message or contact the journal office at ploscompbiol@plos.org. Please include the following items when submitting your revised manuscript:

We look forward to receiving your revised manuscript.

Kind regards,

Patricia M Palagi

Section Editor

PLOS Computational Biology

Russell Schwartz

Section Editor

PLOS Computational Biology

**Journal Requirements:**

3) We note that your Data Availability Statement is currently as follows: "No data involved". Please confirm at this time whether or not your submission contains all raw data required to replicate the results of your study. Authors must share the “minimal data set” for their submission. PLOS defines the minimal data set to consist of the data required to replicate all study findings reported in the article, as well as related metadata and methods (https://journals.plos.org/plosone/s/data-availability#loc-minimal-data-set-definition).

4) Please send a completed 'Competing Interests' statement, including any COIs declared by your co-authors. If you have no competing interests to declare, please state "The authors have declared that no competing interests exist". Otherwise please declare all competing interests beginning with the statement "I have read the journal's policy and the authors of this manuscript have the following competing interests:"

**Reviewers' comments:**

Reviewer's Responses to Questions

**Comments to the Authors:**

Reviewer #1: This piece is generally well written and has the potential to become a valuable guide, but it currently tries to cover too much ground and often misses the level of detail needed to be truly useful. The author should reconsider: who is the intended audience? The text swings between very basic explanations suitable for someone newly introduced to macromolecular structures and highly technical points more suited to someone capable of reprocessing crystallographic data. Narrowing the scope and defining a clear audience would significantly strengthen the piece.

There is also some "lab-speak" creeping in, terms that are familiar to experts but potentially confusing or misleading for beginners. For example, referring to coordinate files as "PDBs" may not be appropriate for those unfamiliar with the nuances of structural data.

In addition, the author should take care to represent the structure and roles of the three wwPDB partner sites (RCSB, PDBe, PDBj) more accurately. While they share a common archive, the tools, interfaces, and data visualisations each site provides are developed independently. A more neutral presentation that acknowledges the work of all three sites would be helpful. It may be worth reaching out to representatives from each to ensure the overview is current and balanced. PDBe, for instance, offers extensive visualisation tools, data aggregation features, and APIs—several of which I have noted where they could be cited but similar detail should be provided for the other sites.

In summary, this is a promising draft with a strong concept behind it, but it would benefit from:

• A clearer sense of the target audience (and consistent level of detail),

• Less reliance on insider jargon, and

• More actionable advice, including direct links to tools, training materials, and relevant publications where appropriate.

Specific pointers are outlined below. Apologies for any repetition with earlier comments—these examples illustrate some (though not all) of the areas where such issues were observed.

Abstract – there are now over 277K structural models

Page 3 second paragraph

“The PDB offers precomputed sequence clusters at certain thresholds, including 100%, 95%, 50%, and 30%, based on the MMseqs2 algorithm”

– the PDB is referred to but the PDB itself is just the archive, it would be more accurate to refer to which website is being used (one or all of RCSB PDB, PDBE or PDBj)

“Many of these selections can be made using the PDB API32, which allows for automated filtering of relevant structural data.”

Here, it would be relevant to be specific that this API is from RCSB PDB or make it more general and cite the APIs of both PDBe and PDBj

It might be worth a sentence in the intro to disambiguate the wwPDB from the different sites

Page 6

“For X-ray crystallography structures, resolution is determined by the diffraction limit of Bragg reflections, defined by the maximum scattering angle at which measurable reflections are observed”

Please add a reference, although, does the audience know what a Bragg reflection is?

“Exploring these metrics can be important”

Strengthen this sentence to be “Exploring these metrics is important”

“The evaluation of experimental data includes checking for twinning in X-ray crystallography data41 or completeness of chemical shift data in NMR42 “

I’m not sure this benefits the audience unless you go into what twinning is or why someone would care – I see this as more data quality and here you want to talk about model fit to data. They are subtly different and for this audience, I would avoid data quality discussions or you will need to greatly expand the discussion on it to make it meaningful.

Random note: why are Q-scores not discussed?

Page 7 paragraph 2 – It is detailed that all these metrics are important to consider but no help is given in knowing what a good model is – either detail this or link to where the reader could discover more. It may be worth referring to the model quality sliders, which are uniform across the PDB member websites or the validation cif and documentation.

This whole section is too light and short for what is needed, the reader is left knowing something has to be reviewed – which is good and the general direction of the assement but ultimately still in the dark as to what to look for.

No mention of Molprobity

Could also mention these https://www.ebi.ac.uk/pdbe/api/doc/validation.html

And also the validation reports

Recommendation 3

I strongly disagree that the level of reader who is reading this guide would be able to 1) reprocess data and 2) make a better job of it than PDBredo.

The space used for recommendation 3 would be better used to expand recommendation 2.

Rec 4

Expand on anisotropic B-factors – you do expand on B factors later but that needs to be here

Recommendation 5

This section is also a list of things to do, but with little help to the reader how to do them or where to find out about them. Who is this guide for?

Recommendation #6:

“Comparing structures with very similar crystallographic parameters” what does this mean exactly? What is ‘very similar’?

“ensuring similar crystallographic properties such as resolution, space group, and unit cells.” This isn’t correct, While it’s true that comparing structures from the same crystal form (e.g., same space group and unit cell) can help isolate biological effects, such matches are rare and not always necessary. Tools like structural alignment can handle different space groups and cell parameters effectively.

The whole paragraph between pages 8-9 needs more thought, it is not clearly written and this is making it misleading. This is the weakest area of the manuscript as it is quiet muddled, it needs to re-thought conceptually.

“If the PDBs come from the same protein” this is a weird turn of phase and a bit science colloquial

“you can compare using chain and residue IDs, but a standard numbering scheme is required. This can be done by manually renumbering chains and residues or employing algorithms such as PDBRenum to map PDB residue numbers onto UniProt numbering, which will also allow integration with other databases (see Recommendation #9)75. If PDBs are related, you can also align them based on a multiple sequence alignment.”

You mention SIFTS before (and then in the next section) but not here, why? Maybe this all should be brough together.

It would also be good to mention the PDBe-KB protein pages for structural comparison and clustering

https://www.ebi.ac.uk/pdbe/pdbe-kb/

https://www.ebi.ac.uk/pdbe/pdbe-kb/proteins/2etx

https://pubmed.ncbi.nlm.nih.gov/34755867/

In section 9, page 10

Would be worth mentioning the PDBe ligand pages

https://www.ebi.ac.uk/pdbe-srv/pdbechem/chemicalCompound/show/STI

https://pubmed.ncbi.nlm.nih.gov/40100137/

And PDBe CCDUtils

https://pubmed.ncbi.nlm.nih.gov/38042830/

“However, users must remain mindful of the “last-AÅngstrom” problem, “

What does this mean exactly? It is not explained anywhere and is not a common phrase

Reviewer #2: The author gives a short review-style overview of how to deal with structure models from the PDB. Although it is not a unique overview (there have for instance been many book chapters on the topic), it is a good read for anyone who is relatively new to structural bioinformatics. It is easy to read and covers all the big issues in dealing with PDB entries. I do have some notes that may improve the manuscript:

• Introduction, first paragraph: Please rephrase the first sentence. Helen Berman has done a lot for the PDB over many years, but she was not in the lead when the PDB was founded.

• Introduction, first paragraph: The PDB does not contain structures but models thereof. Please use ‘structure models’ or ‘entries’.

• Introduction, first paragraph: The PDB is not only robust, it was also the first (still active) open-access resource in biology.

• Page 3, first line: Please elaborate a bit on the symmetry-related protomers. These can be purely coincidental with non-crystallographic symmetry, part of a homo-multimer, or part of a homomultimer that also needs application of crystallographic symmetry. Explain that you can download biological assemblies from the PDB.

• P3, end of Recommendation #1: The mention API is not the PDB API, but rather the RCSB-API. Please also mention that PDBe has a very rich API. I don’t know about PDBj, but the author should look into that.

• P3, last paragraph: NMR entries do not a have a reported resolution.

• P4, 2nd paragraph: The PDB doesn’t mandate anything in terms of models having a minimal quality. It just test for the quality with is validation pipeline and make those results public. You can essentially still deposit rubbish. Please refer to the papers from the different validation taskforces from the wwPDB such as Read, et al. (see: https://www.wwpdb.org/task/validation-task-forces).

• P4, 2nd paragraph: AFAIK the twinning evaluation in the PDB validation server is done by Xtriage which should probably be cited. Density validation things originally all came from the Electron Density Server (Kleywegt et al., 2004). Currently the RS is calculated by the program density-fitness (van Beusekom et al, 2018). See https://www.wwpdb.org/validation/XrayValidationReportHelp

• P5, first paragraph: Please also mention DNATCO which is a validation to for DNA/RNA backbones.

• Recommendation #3: I think it is written as ‘PDB-REDO’ nowadays.

• Recommendation #3: MTZ is a data format. The recommendation in 1998 was about deposition of reflection data, not about the format. Please also note that deposition of reflection data became mandatory in 2008. There are roughly 10k crystallographic PDB entries without reflection data.

• P6, 2nd paragraph: The team behind PDB-REDO published a paper about backfilling missing loops by homology with a program called Loopwhole. Please have a look at that.

• Page 6, end of Recommendation #4: I couldn’t agree more!

• P7, 2nd par: there is a reference to #5 in #5.

• Recommendation #9: Please mention the PDBe-KB which is an important resource in this context. Perhaps also mention the 3D-Beacons network that connects many structural biology resources.

• P11, first paragraph: In the context of AlphaFold and missing ligands and co-factors, the author could point to AlphaFill which tries to fill this void.

Reviewer #3: This is a useful opinion piece on how to conduct structural bioinformatics studies. It follows a long series of "Ten Simple Rules" pieces published in PLOS CB. It is well written and quite clear. It provides suitable guidance for people new to structural bioinformatics techniques.

I just have a few things that could be added, expanded, or emphasized.

1) The distinction between asymmetric units and biological assemblies should be described. The PDB provides separate mmCIF and PDB-format files for biological assemblies. Usually these come from the authors, but some of them are also generated by software (PISA). The latter occurs when the PISA assembly is larger than the authors' biological assembly. For most scientific studies, only the biological assemblies should be used. The ms could warn readers that about 20% of the assemblies in the PDB are incorrect. The ProtCID and ProtCAD databases are useful sources for sorting out true from false assemblies.

2) In a similar vein, the author describes the importance of crystal parameters in separating structures. This is the idea of "crystal forms" used in our ProtCID and ProtCAD databases. The idea is that to understand protein assemblies, it is the number of unique crystal forms that is truly informative (e.g. on identifying the correct homodimer for a set of homologous family of proteins). This is also applicable for studying loop conformations that may be in crystal contacts. Understanding and grouping a set of structures into crystal forms can allow someone to analyze loop conformations in different crystallographic contexts in order to eliminate the confounding role of crystallization artifacts.

3) I think a few words are in order for the efforts of Helen Berman and colleagues in remediating the PDB data after they took over from Brookhaven in 1998. These efforts are the absolute foundation of our being able to perform structural bioinformatics. There are several papers from Berman et al that could be cited. The effort is ongoing and includes efforts by PDBe and PDBj.

4) I think there could be a short section on the value of the mmCIF format. It is much richer than PDB format. It is necessary for structures with new ligands, which now have 5 characters and therefore are not available in legacy PDB format. Neither are large structures. An imporant feature of mmCIF is that it contains a mapping of the protein sequence to the residue numbering in the coordinates. That is, one could number a protein chain from 1 to the length of the chain. But authors usually use a different numbering (e.g. from the Uniprot sequence). Different tables in the mmCIF files use either the author numbering or the 1-to-N numbering or both. Our PDBrenum database renumbers the author numbering in all tables so it agrees with the Uniprot numbers (it will be updated very soon and should be up-to-date going forward).

5) We find the ECOD database to be very useful. for instance in comparing assignments to AFDB structures, TED/CATH is missing many domains altogether, while ECOD has most of them. Same for Zheng et al(2017) on p. 4.

6) very minor: p.9: the SIFTS reference is given as "Velankar et al(2013)" instead of being a numbered reference.

7) very minor: p.2: "whether your protein are part of large complexes"  "whether your protein is part of large complexes"

**Have the authors made all data and (if applicable) computational code underlying the findings in their manuscript fully available?**

Reviewer #1: Yes

Reviewer #2: Yes

Reviewer #3: Yes

PLOS authors have the option to publish the peer review history of their article (what does this mean? ). If published, this will include your full peer review and any attached files.

**Do you want your identity to be public for this peer review?** For information about this choice, including consent withdrawal, please see our Privacy Policy .

Reviewer #1: No

Reviewer #2: No

Reviewer #3: **Yes: ** Roland Dunbrack

**Figure resubmission:**

**Reproducibility:**



---

## [Decision Letter · Decision Letter 1]

3 Oct 2025

PCOMPBIOL-D-25-00823R1

10 Rules for a Structural Bioinformatic Analysis

PLOS Computational Biology

Dear Dr. Wankowicz,

Thank you for submitting your manuscript to PLOS Computational Biology. After careful consideration, we feel that it has merit but does not fully meet PLOS Computational Biology's publication criteria as it currently stands. Therefore, we invite you to submit a revised version of the manuscript that addresses the points raised during the review process.

As soon as you have dealt with the comments of Reviewer #2, I will do a quick review myself and then proceed to the next steps. Thank you for addressing these minor but relevant comments. 

Please submit your revised manuscript within 30 days Dec 03 2025 11:59PM. If you will need more time than this to complete your revisions, please reply to this message or contact the journal office at ploscompbiol@plos.org. Please include the following items when submitting your revised manuscript:

We look forward to receiving your revised manuscript.

Kind regards,

Patricia M Palagi

Section Editor

PLOS Computational Biology

**Reviewers' comments:**

Reviewer's Responses to Questions

**Comments to the Authors:**

Reviewer #1: This is much improved and more consistent; this will certainly be a handy guide. One small note: you have a type here "It is also important to note that in borh X-ray and cryo-EM"

Reviewer #2: I enjoyed reading this much improved version of the manuscript. I recommend it for publication with only a few minor corrections that, in my opinion do not require additional review:

- The number of released PDB entries is just over 242k, please correct the number in the manuscript. Not sure where the 277k came from, this might reflect the number of entries including things that are not released yet.

- Please rephrase the sentence "By democratizing...novel discoveries" in the abstract.

- EM maps show electron potential rather than density. This should be corrected in several spots in the manuscript, e.g. by just using 'map' instead of 'density'.

- Recommendation 2: NMR entries do not have a reported resolution.

- Page 4, last sentence of first paragraph: It is also important to note that in boTh X-ray and cryo-EM, while two structures (or rather their underlying data) can have the same resolution, they can be modeled to different levels of ACCURACY, .....

- Page 6, first paragraph: the statement of missing loops is somewhat in conflict with what the author says about Loopwhole in #2. Perhaps 'correct them' -> 'correct all of them'?

- No need to add it at this stage, but on page 8 it would have been good to mention alignment free structure comparison. This can be done in torsion space (or an encoding thereof) or by using local distances such as with ProSMART or hydrogen bonds as done by HODER.

Reviewer #3: The author has responded to my questions and those of the other reviewers very well. The paper is ready to be accepted.

**Have the authors made all data and (if applicable) computational code underlying the findings in their manuscript fully available?**

Reviewer #1: Yes

Reviewer #2: Yes

Reviewer #3: Yes

PLOS authors have the option to publish the peer review history of their article (what does this mean? ). If published, this will include your full peer review and any attached files.

**Do you want your identity to be public for this peer review?** For information about this choice, including consent withdrawal, please see our Privacy Policy .

Reviewer #1: No

Reviewer #2: No

Reviewer #3: No

**Figure resubmission:**
---

## [Editor Report · Decision Letter 2]

9 Oct 2025

Dear Dr Wankowicz,

We are pleased to inform you that your manuscript '10 Rules for a Structural Bioinformatic Analysis' has been provisionally accepted for publication in PLOS Computational Biology.

Best regards,

Patricia M Palagi, Russell Schwartz, Scott Markel and Francis Ouellette

Section Editor

PLOS Computational Biology

---

## [Editor Report · Acceptance letter]

PCOMPBIOL-D-25-00823R2

10 Rules for a Structural Bioinformatic Analysis

Dear Dr Wankowicz,

I am pleased to inform you that your manuscript has been formally accepted for publication in PLOS Computational Biology. Your manuscript is now with our production department and you will be notified of the publication date in due course.

With kind regards,

Anita Estes
